# Evaluation of the Dietary Arginine Supplementation on Yellow Catfish: From a Low-Temperature Farming Perspective

**DOI:** 10.3390/biology13110881

**Published:** 2024-10-29

**Authors:** Quanquan Cao, Mohamed S. Kisha, Alkhateib Gaafar, Abdelgayed Metwaly Younes, Haifeng Liu, Jun Jiang

**Affiliations:** 1College of Animal Science and Technology, Sichuan Agricultural University, Chengdu 611130, China; 2Hormones Department, National Research Centre, Dokki, Cairo 12622, Egypt; kishtamsa@gmail.com; 3Hydrobiology Department, National Research Centre, Cairo 999060, Egypt; alkhateibyg@yahoo.com (A.G.); tahoon176@yahoo.com (A.M.Y.)

**Keywords:** yellow catfish, arginine, growth performance, digestive ability, antioxidant ability, low-temperature

## Abstract

The study focused on the effects of dietary arginine (Arg) supplementation on yellow catfish (*Pelteobagrus fulvidraco*) under low-temperature stress (18 °C). Over eight weeks, 720 fish were divided into six groups and fed diets with varying Arg levels. The results showed that Arg supplementation improved growth, feed efficiency, digestive enzyme activity, and antioxidant responses in the fish. It also enhanced the expression of antioxidant-related genes. Based on the findings, the optimal Arg intake for yellow catfish under these conditions is 26.8 g per kg of diet, equivalent to 37.0 g of dietary protein.

## 1. Introduction

There is evidence suggesting that environmental stress can disrupt homeostasis and negatively impact the functioning of biological systems [1]. Water temperature is commonly recognized as one of the significant factors influencing aquaculture. Accumulating evidence has demonstrated that low temperatures decrease growth, food intake, and feeding efficiency [2,3,4,5,6]. Previous studies have shown that dietary phosphorus supplementation under low temperature improves cell viability, antioxidative capacity, energy production, and lipid transportation in puffer fish [7], suggesting that specific nutrients may play a significant part in ameliorating sub-low temperature-induced injury. However, limited research has been conducted on the effects of nutrients on fish exposed to sub-low temperature conditions.

Arginine (Arg), an indispensable amino acid for fish [8,9], plays multiple roles in cell metabolism and regulation. Apart from its requirement for protein synthesis, Arg is also involved in synthesizing important biochemicals such as ornithine, polyamines, proline, creatine, and nitric oxide [10]. Recent studies have increased on the dietary Arg requirements in different fish species. For juvenile yellow groupers (*Epinephelus awoara*) at 27.5 °C, the optimal amount of dietary Arg is estimated to be 2.8% of the dry diet [11], 2.17% for grass carp (*Ctenopharyngodon idellus*) at 28 °C [12], 2.85% for juvenile cobia (Rachycentron canadum) at 30 °C [13], 1.84% for blunt snout bream (*Megalobrama amblycephala*) at 30 °C [14], 2.46% for Indian major carp (*Cirrhinus mrigala*) at 26.5 °C [15], and 1.80% for Jian carp (*Cyprinus carpio*) at 26 °C [16]. It has been reported that a dietary requirement of 2.38% arginine (Arg) in dry diet is necessary for optimal growth of juvenile yellow catfish (*Pelteobagrus fulvidraco*) at 29 °C [17]. Fish have varying Arg requirements at different life stages, growth stages, physiological stages, and environmental conditions, such as water temperature. However, there is limited information available regarding the Arg requirements of yellow catfish at cold temperatures. Fish rely on digestive enzymes and brush border enzymes to digest and absorb nutrients for growth [18]. Several digestive enzymes, including trypsin, chymotrypsin, lipase, and amylase, are produced in the fish exocrine pancreas [19]. Other enzymes, such as γ-glutamyl transpeptidase (γ-GT), alkaline phosphatase (AKP), Na^+^/K^+^-ATPase (NKA), and creatine kinase (CK), are essential for the final stages of food digestion and assimilation in fish [20]. In a previous study, dietary supplementation of Arg improved the digestive and absorptive abilities of carp (*Cyprinus carpio*) [21]. Similarly, in weaned piglets, the addition of Arg to the diet promoted the development of intestinal mucosa and improved the activity of intestinal sugar digestive enzymes [22]. However, there is currently no research available on the influence of Arg on the digestive abilities and absorptive functions of yellow catfish, particularly under sub-low temperature, which necessitates further investigation.

The intestine plays a crucial role as the primary site for antioxidant capacity in fish [23]. The maintenance of structural integrity and functionality in the fish intestine relies on the capacity of the intestinal antioxidant system [24]. Oxidative damage occurs when the rate of reactive oxygen species (ROS) scavenging by the antioxidant system is lower than the rate of ROS production, as noted by Matés and Sánchez-Jiménez [25]. Li et al. (2014) found that shrimp exhibited increased ROS content when exposed to cold temperatures (17 °C) [26]. Previous research indicates that low water temperatures induce cytomembrane oxidation and antioxidative actions, resulting in increased malondialdehyde (MDA) and protein carbonyl (PC) contents in tail skeletal muscle [27]. Low temperatures have also been found to decrease antioxidant enzyme activities in *Litopenaues vannamei* [4], suggesting that low temperatures reduce antioxidant capacity. Fish intestinal antioxidant capacity improves with enhanced enzyme activities, increased non-enzymatic antioxidant levels, and upregulated antioxidant enzyme gene expression. Nuclear factor-red lineage 2-related factor-2 (Nrf2) is well characterized in sensing oxidants and regulating the expression of antioxidant enzymes-related genes [28]. The addition of Arg to the diet of grass carp increased the intestinal antioxidant enzymes activities and non-enzymatic antioxidant content. However, it remains uncertain whether dietary Arg supplementation can decrease low temperature-induced oxidative damage by enhancing the antioxidant capacity of the intestine.

Yellow catfish has gained popularity in China’s freshwater fish farming industry due to its delicious meat quality and high economic value [29]. Being a poikilothermic fish, the water temperature plays a crucial role in its life [30]. The yellow catfish usually grows in freshwater between 20 °C and 32 °C. The optimum growth temperature is 26.8 °C and temperatures below 19 °C leads to reduce food intake and growth performance in yellow catfish [31]. In China, yellow catfish experience their last growth stage during fall when the water temperature drops to 18 ± 2 °C. However, yellow catfish farming faces challenges in sub-low temperature environments, which can negatively impact fish health and growth. Addressing these challenges is crucial to ensure sustainable aquaculture production. One potential solution lies in optimizing the nutritional composition of fish diets, specifically through dietary supplementation of arginine. The dietary requirements of Arg for yellow catfish under these conditions are not yet clearly understood. This study aims to address the critical need for optimized dietary formulations for yellow catfish, particularly in low-temperature farming conditions where growth rates and feed efficiency can be severely compromised. By evaluating the effects of dietary arginine supplementation, this research seeks to enhance the understanding of amino acid requirements in yellow catfish, ultimately contributing to more sustainable aquaculture practices in challenging environments.

## 2. Materials and Methods

### 2.1. Experimental Diets

The L-Arg (>99% purity) used in this study was supplied by Hengyuan Biotech Co., (Shanghai, China). The main protein sources for the test diets included fish meal, rapeseed meal, soybean meal, and maize protein meal, providing a protein content of approximately 37% (Table 1). The diets contained approximately 7% lipid, with soya bean oil being the main lipid source. Six test diets were prepared by adding L-Arg to the diet at concentrations of 1.79 g/kg (control), 2.11 g/kg, 2.36 g/kg, 2.68 g/kg, 2.95 g/kg, and 3.26 g/kg, respectively. All feeds were isonitrogenous and isoenergetic by the addition of glycine. The granular mixture was extruded using a 2 mm diameter screw extruder (SYX62, Xiamen, China), air-dried, and stored in a refrigerator at −20 °C.

### 2.2. Fish Trial

Yellow catfish were obtained from Sichuan Rongsen Corporation (Sichuan, China) and acclimatized to the laboratory environment for one month. In the experiment, 720 yellow catfish, averaging 61.2 ± 0.3 g in initial weight, were randomly distributed across 18 tanks (200 cm × 100 cm × 105 cm), with 40 fish per tank. They were fed their respective diets twice daily for 8 weeks to ensure satiety. The water temperature was measured three times a day (08:00, 13:00, and 18:00), and the average water temperature was maintained at 18 ± 2 °C (Automatic control equipment tank, Qingdao Zhongyi Limited Liability Company). Other water quality parameters were measured at 8:00 daily, with a pH of 7.0 ± 0.3 and dissolved oxygen levels maintained above 5 mg/L. A continuous flow rate of 1.2 L/min per water tank was maintained. The photoperiod was natural and all tanks had similar lighting conditions. The study received approval from the Animal Care Advisory Committee of Sichuan Agricultural University under permit No. DKY-S20170512.

### 2.3. Sample Collection and Analysis

Any uneaten feed was collected 30 min after each feeding session and subsequently dried in an oven at 65 °C to determine the feed intake (FI). Initial body weight (IBW) and Final body weight (FBW) were measured and survival rate (SR), percent weight gain (PWG), specific growth rate (SGR) and feeding efficiency (FE) were calculated. Fish in each tank were weighed at the start and conclusion of the feeding trial. Four fish per tank were randomly selected to assess carcass proximate composition. Twenty-four hours after the final feeding, 12 fish from each group were anesthetized with tricaine methane sulfonate (MS222) at 120 mg/L. Samples were collected to measure the following: visceral index, stomach protein content (SPC), pancreatic protein content (PPC), intestinal protein content (IPC), liver protein content (LPC), hepatosomatic index (HSI) and intestinal enzyme activities [32]. Additionally, twelve additional fish were removed from each tank. Their intestines were quickly extracted, kept in liquid nitrogen, and preserved at −80 °C until gene expression was determined.

### 2.4. Biochemical Analysis

The dry matter content of the diets and fish were analyzed according to the method 934.01, (AOAC, 1990). The samples were desiccated at a temperature of 104 °C until constant weight for the analysis of dry matter. The total nitrogen (N) was determined according to the method 968.0 (AOAC, 1990) after acid digestion was performed. The crude protein (CP) was obtained by multiplying the total N content by 6.25. The moisture was determined according to the method 935.29 (AOAC, 1990). The crude lipid (EE) was analyzed using the Soxhlet extraction method 983.15 (AOAC, 1990), and the ash contents were examined in the muffle furnace at 550 °C for 4 h according to the method 942.05.

The method used to determine the approximate composition of experimental feeds and entire carcasses was established by the Association of Official Analytical Chemists (AOAC, 1990). Pancreatic, gastric, and intestinal samples were homogenized in a ratio of 1:10 (*w*/*v*) in ice-cold saline and then centrifuged at 6000× *g* at 4 °C for 20 min. Enzyme activity and protein content were determined from the supernatant. The tissue protein contents of each group were measured based on the Bradford method [33]. The activities of pepsin, trypsin, and chymotrypsin were determined according to [34]. Amylase and lipase enzyme activities were measured using the method described by. AKP, NKA, γ-GT, and CK activities were measured as outlined by [35].

The MDA and PC contents were determined using the method described by [36]. CAT activity was measured according to the method [37]. SOD and GPX activities were determined as outlined by [38]. GST activity was measured by monitoring the decrease in GSH concentration, and GR activity was determined by measuring the depletion of NADPH during GSH production according to [39]. The total glutathione (T-GSH) content was measured according to [40], while the anti-superoxide anion (ASA) and anti-hydroxyl radical (AHR) capacity were measured according to [41].

### 2.5. Real-Time Quantitative PCR

Real-time quantitative PCR was used to detect gene expression. Total RNA was extracted from the intestine using TRIZOL reagent (Invitrogen, Carlsbad, CA, USA) according to the manufacturer’s instructions. RNA quantity and quality were assessed by 1% agarose gel electrophoresis and spectrophotometry at 260 and 280 nm. cDNA synthesis was performed with a PrimeScript^®^ Reverse Transcription Kit and gDNA Eraser (TaKaRa, TaKaRa Bio Inc., Shiga, Japan). Specific primers for CuZnSOD, CAT, GPX1a, GCLC, and Nrf2 genes were designed using Primer Premier software (Premier Biosoft International, Palo Alto, CA, USA) based on yellow catfish sequences (GenBank accession nos. KX455916, KX455919, KY312111, KX455918, KX455917) (Table 2). Real-time PCR analysis was conducted with the CFX96 Real-Time PCR Detection System (Bio-Rad, Hercules, CA, USA). The PCR conditions included initial denaturation at 98 °C for 1 min, followed by 40 cycles of denaturation at 95 °C for 5 s and annealing at the temperature specified in Table 2 for 30 s. PCR efficiency was estimated using cDNA sequence dilution analysis, aiming for near 100% efficiency. Relative gene expression was calculated using the 2^−ΔΔCT^ method was used to calculate relative gene expression. To normalize the cDNA load, β-actin was utilized as the internal reference gene. The relative expression of the target gene was then normalized to the expression of β-actin in the same sample.

### 2.6. Data Analysis

Data were analyzed using the PROC MIXED procedure of SAS 9.3 (SAS Inst. Inc., Cary, NC, USA) with the model: *Y_ijkl_* = *μ* + *T_i_* + *P_j_* + *S_k_* + *C*(*_k_*)*_l_* + *T* × *S_ik_* + *e_ijkl_*, where *Y_ijkl_* refers to the dependent variable; *μ*, the overall mean; *T_i_*, the fixed treatment effect; *P_j_*, the random period effect; *S_k_*, the random square effect; *C*(*_k_*)*_l_*, the random effect of the *l*th steer within the *k*th square; *T* × *S_ik_*, the interaction between the *i*th treatment and the *k*th square; and *e_ijkl_*, the error residual. The linear and quadratic effects of increasing levels of GA were evaluated using the CONTRAST procedure of SAS 9.3. The Kenward-Roger option was used to calculate the degrees of freedom. Differences among the means of different treatments were tested using Duncan’s test. Effects were considered significant at *p* ≤ 0.05. All values are presented as least squares means ± standard errors (SE).

## 3. Results

The mortality rate was low, with only one death occurring among all the groups fed with 2.11% dry feed after one week.

### 3.1. Growth Performance and Somatic Parameters

The parameters of PWG, SGR, FE, FI, and PER showed a significant increase when the Arg level reached 2.68 g/kg, followed by a significant decrease, as presented in Table 3 (*p* < 0.05). The SGR was lowest in the basal diet group (*p* < 0.05). Based on the quadratic regression equation for SGR, the Arg requirement for yellow catfish was evaluated as 2.68 g/kg diet in the fall (water temperature: 18 ± 2 °C), corresponding to a diet protein content of 37 g/kg (Figure 1).

Table 4 presents the net nutrient deposition and whole fish compositions for yellow catfish. At Arg level 2.68 g/kg, the protein content of the fish increased significantly (*p* < 0.05) and remained relatively constant thereafter (*p* > 0.05). The lipid content gradually increased with the addition of Arg up to 2.68 g/kg, reached a maximum, and then gradually decreased (*p* < 0.05). Similar changes were observed in the levels of PPV and LPV with increasing lipid content. There were no significant changes in water content, ash content, and maximum water content among different groups (*p* > 0.05). In the 2.11 g (Arg/kg) diet, the small intestinal length (RGL) was the largest, while in the control group, the small intestinal length (RGL) was the smallest (Table 5, *p* < 0.05). Pancreatic, intestinal, and liver protein content increased significantly at dietary Arg levels of 2.36 and 2.68 g/kg, respectively, before gradually decreasing (Table 5, *p* < 0.05). The liver body index (HSI), intestinal body index (ISI), and gastric protein content (SPC) did not show significant differences between dietary groups (*p* < 0.05).

### 3.2. Enzyme Activities

Table 6 displays the digestive enzyme activities of yellow catfish. Trypsin, chymotrypsin, and pancreatic lipase activities significantly increased (*p* < 0.05) with increasing levels of Arg up to 2.68 g/kg in the diet. The highest pancreatic amylase activity was observed at 2.36 g Arg/kg diet. When Arg levels in the diet reached 2.68, 2.36, and 2.95 g/kg, respectively, pancreatic amylase, chymotrypsin, and lipase activities in the intestine increased significantly and then decreased (*p* < 0.05). Intestinal amylase activity did not change significantly. The γ-GT activity of PI and DI increased significantly with increasing Arg levels at 2.36 and 2.68 g/kg diet, respectively, followed by a decreasing trend (Table 7). The group with 2.95 g/kg Arg showed the highest CK activity in PI, while the group with 3.26 g Arg/kg exhibited the lowest CK activity (*p* < 0.05). Creatine kinase activity increased significantly with increasing Arg amount until reaching 2.36 g/kg, after which it decreased rapidly (*p* < 0.05). The addition of 2.68 g Arg/kg to the diet resulted in the highest NKA activity in PI and DI. The highest AKP activity in PI and DI was observed in the group with 2.36 g Arg/kg added to the diet, while the lowest was in the group with 3.26 g Arg/kg (*p* < 0.05). The following equation was obtained for AKP activity in DI: *Y* = −23.604X^2^ + 117.556X − 120.526, *R*^2^ = 0.751, *p* < 0.05.

### 3.3. Intestinal Antioxidant Parameters of Yellow Catfish

Table 8 presents the effect of dietary Arg on intestinal antioxidant activity in yellow catfish. At a dietary Arg level of 2.68 g/kg, both MDA and PC levels decreased significantly (*p* < 0.05) as the dietary Arg level increased, but then rose again with further increases in Arg. T-SOD and GPX activity increased significantly (*p* < 0.05) at 2.68 and 2.36 g/kg diets, respectively, and remained stable thereafter (*p* > 0.05). CAT, GST, and GR activities increased significantly with rising Arg levels (*p* < 0.05), up to a maximum at 2.68 g/kg, after which they decreased significantly (*p* < 0.05). T-GSH content was the highest at 2.38 g Arg/kg feed and the lowest at 3.26 g Arg/kg (*p* < 0.05). The ASA exhibited a similar trend to T-SOD. AHR content significantly declined as dietary Arg levels exceeded 2.68 g/kg (*p* < 0.05). Figure 2 and Figure 3 display the relative gene expression of CuZnSOD, CAT, GPX1a, GCLC, and Nrf2 in the intestine of yellow catfish at different Arg levels. As Arg increased to 2.11, 2.36, and 2.68 g/kg in the diets, the expression levels of CuZnSOD, CAT, and GPX1a significantly increased (*p* < 0.05) and then stabilized (*p* > 0.05). In the 2.95 g/kg diet, GCLC mRNA abundance was significantly up-regulated (*p* < 0.05) and subsequently decreased as Arg levels increased (*p* < 0.05). Dietary Arg significantly up-regulated the expression level of intestinal Nrf2, with the highest expression observed in the 2.68 g/kg Arg diet group (*p* < 0.05).

### 3.4. Principal Component Analysis (PCA)

PCAs were conducted using gene expressions (GCLC, GPX1a, Nrf2, CAT, CuZnSOD) in the intestine (Figure 4). Principal Component 1 (PC1) described 64.5% of the original information, and PC2 described 16.8%, resulting in a cumulative percentage of 81.3% in the gut. A comparison between PC1 and PC2 was performed to understand the contribution of the main components. In the biplot, GCLC, Nrf2, and GPX1a of the intestine formed a positive load on the right side, while CAT and CuZnSOD formed a negative load on the right side. The same variables were used to plot PCA individual factor maps, expressing each factor as the same variable. Both groups of fish were located on the left side of the control area, with 1.79 and 3.26 g/kg Arg as the control. There was a significant population distribution along the PC1 axis in the intestine. In the 2.11, 2.36, and 2.68 g/kg Arg dry diets, fish appeared on the positive side of the biplot along PC2 (Figure 4).

## 4. Discussion

### 4.1. Effect of Dietary Arg Level on Growth in Sub-Low Temperature Environments

This study investigated the impact of dietary Arg on the growth of yellow catfish under specific temperature conditions. Yellow catfish reared at 18 °C required 2.68 g/kg Arg in the diet to achieve maximum weight gain and a better specific growth rate (SGR). This requirement is higher than the 2.38 g/kg Arg diet observed in yellow catfish juveniles at the optimal water temperature of 29 °C [42]. However, the SGR in this study was lower than that of juvenile yellow catfish in previous studies [42], consistent with findings in juvenile yellow catfish [43], and juvenile common carp [44], where SGR was lower due to water temperature. Water temperature is an important factor influencing the growth process of fish [45]. In nature, fish attempt to adjust themselves to temperature fluctuations, which can impact their overall physiology and result in lower growth rates. This study also demonstrated that SGR increased with the increasing Arg content in the diet, up to an optimum level (2.268 g/kg), which has been observed in Indian major carp [46,47], black sea bream [48], and rainbow trout [30]. The feed intake (FI) and feed efficiency (FE) also improved in response to increasing levels of Arg (Arg), This result shows that the increase in fish growth is partially attributed to the increase in FI and FE. According to [46], growth is the result of high protein deposition due to the intake of essential amino acids. Our study found a significant increase in the protein retention rate (PRV) when maintaining the optimum concentration of Arg for specific growth rate (SGR). Additionally, Arg has a significant secretory function, which is related to its impact on fish growth and development [49].

### 4.2. Effect of Dietary Arg Level on Digestive and Absorptive Enzyme Activities

The effect of dietary Arg level on digestive and absorptive enzyme activities is a key factor affecting fish growth and development [50]. It depends on the enzyme activity of the digestive and brush border membrane, responsible for the degradation and absorption of food [51]. Not only can water temperature affect cell membrane structure [52], but it can also have an impact on the digestive process [53,54]. Understanding the enzyme dynamics and digestive processes under seasonal temperature changes is crucial for preparing feeds that optimize growth performance and reduce feed costs for fish. The pancreas of yellow catfish can secrete pancreatic juice containing trypsin, amylase, chymotrypsin, and lipase of the intestine independently of the liver. Many transport systems, such as the entry of amino acids, phosphate, or glucose into cells, utilize the sodium gradient potential energy generated by the sodium-potassium adenosine triphosphatase (NKA) [55]. Alkaline phosphatase (AKP) is involved in the uptake of nutrients like glucose, calcium, lipids, and inorganic phosphates [56]. Gamma-glutamyltransferase (γ-GT) plays an essential role in the active transport of amino acids [57]. Creatine kinase (CK) has a key role in cellular energy metabolism as it catalyzes the transfer of phosphate to creatine in an ATP-dependent manner. When the dietary Arg level increased, there was a significant increase in gastric pepsin activity, as well as hepatopancreatic and intestinal trypsin, chymotrypsin, and amylase activities. Moreover, increasing levels of dietary Arg also led to a significant increase in intestinal NKA, AKP, γ-GT, and CK activities. It is evident that Arg improves digestion and absorption in fish under low-temperature conditions. To the best of our knowledge, this is the first study on the influence of Arg on the digestive and absorption enzyme activities of yellow catfish. Similar results were obtained in Jian carp at the optimum temperature [14]. A study on bull’s-eye pufferfish showed a correlation between digestive enzyme activity and the development of the digestive tract [55]. Thus, the addition of appropriate levels of Arg at low temperatures can promote the development of the hepatopancreas and intestinal tract of yellow catfish, thereby increasing digestive enzyme activity. In our research, Arg significantly improved pancreatic protease capacity (PPC), relative gut length (RGL), and intestinal protease capacity (IPC). Furthermore, Arg can be metabolized by nitric oxide synthase (NOS) to form NO, and by arginase and Arg decarboxylase to produce spermine. Previous studies have demonstrated that NO can stimulate pancreatic secretion in pigs [58] and spermine can increase enzyme activity in the pancreas of black bass *Dicentrarchus labrax* [59]. Therefore, it is hypothesized that Arg could improve the secretion of digestive enzymes by stimulating NO and spermine in yellow catfish, which warrants further investigation.

### 4.3. Effect of Dietary Arg Level on Intestinal Anti-Oxidative Capacity

Reactive oxygen species (ROS) are chemically reactive molecules containing oxygen that are natural by-products of mitochondrial oxygen metabolism and play a vital role in cell signaling and homeostasis [60]. When the generation and accumulation of ROS exceed the body’s ability to counteract these reactive substances, several biomolecules, including lipids and proteins, undergo oxidative damage [61]. Li et al. (2014) reported that exposure of shrimp to low temperatures increased the levels of ROS [26]. Similarly, Kake-Guena et al. (2017) reported that temperature changes affect mitochondrial function, leading to an increase in ROS formation [27]. low water temperatures (15 °C and 21 °C) induce stress scenarios that can overwhelm the antioxidant system, resulting in overproduction of ROS. As a result, the addition of Arg to the diet at low temperatures (18 °C) has been shown to reduce malondialdehyde (MDA) and protein carbonyl (PC) levels in the intestine [4]. The similar results have been shown in intestine of grass carp [32]. Therefore, Arg could prevent oxidative stress in the intestine of yellow catfish under seasonal cold stress. Recent studies have found that antioxidant enzymes are essential for protecting cells from oxidative stress [62]. Superoxide dismutase (SOD) is an enzyme that is the first to respond to O_2_ radicals and serves as an essential endogenous antioxidant protecting against oxidative stress. Catalase (CAT) is also essential for defending cells against the toxicity of hydroxyl radicals, while glutathione-S-transferase (GST), glutathione peroxidase (GPx), and glutathione reductase (GR) are GSH-dependent enzymes that counteract peroxidative damage [63]. In a recent study conducted at cold temperatures, the addition of 2.68 g of Arg per kg of diet increased the enzymatic activities of CuZnSOD, CAT, GST, and GPX in the fish intestine. This increase in enzyme activity can be attributed to the enhanced synthesis of enzyme proteins, which is linked to gene transcription and translation [64]. Similarly, previous studies have indicated that dietary addition of Arg enhances tissue antioxidant status and reduces protein and fat oxidation in various animals. The results of the present study indicate that CuZnSOD, CAT, and GPX1a showed similar patterns of change in mRNA levels and enzymatic activity in the intestine. Similar findings have been reported for Arg supplementation in pig [41] and grass carp [65,66]. GSH protects cells from oxidative stress by serving as the major endogenous antioxidant scavenger. GSH/GSSG represents the main cellular redox buffer [67]. Glutamate cysteine ligase (GCL) has been identified as the rate-limiting enzyme for glutamate (Glu) biosynthesis [68], and GSH synthase is coordinately regulated as a subunit of glutamate cysteine ligase. The present study indicated that dietary supplementation of Arg increased intestinal GSH content and the transcriptional level of GCLC. This suggests that Arg supplementation could promote GSH synthesis by upregulating GCLC expression under cold temperatures [68].

Nrf2 is a crucial transcription factor that regulates the expression of antioxidant enzyme genes in fish. Its facilitated translocation into the nucleus plays an essential role in upregulating antioxidant enzyme gene expression [28,46]. The current study indicated that optimal levels of Arg (2.268 g/kg) resulted in the upregulation of intestinal Nrf2 mRNA expression levels in yellow catfish, exhibiting a similar pattern to the expression levels of CuZnSOD, CAT, and GPX mRNA [26]. This suggests that dietary Arg supplementation could enhance the expression of CuZnSOD, CAT, and GPX genes by upregulating Nrf2 mRNA expression. However, further investigation is needed to understand the potential mechanism (Figure 5).

## 5. Conclusions

This study found that dietary supplementation of Arg improved the growth performance, digestion and absorption capacity, and intestinal antioxidant status of yellow catfish in sub-low temperature environments. The optimal Arg level is 2.68 g/kg. It also increased antioxidant capacity by upregulating Nrf2 mRNA expression and the expression of antioxidant-related genes. Quadratic regression analysis was employed to estimate the dietary Arg requirement for yellow catfish at 18 °C, which was determined to be between 61.0 g and 89.0 g, based on a dry feed specific growth rate (SGR) of 2.68 g/kg and a feed protein content of 37.0 g/kg. These findings suggest that optimizing the dietary arginine content in yellow catfish feeds can contribute to better pancreas health, improved intestinal development, and overall fish performance in sub-low temperature aquaculture systems.

## Figures and Tables

**Figure 1 biology-13-00881-f001:**
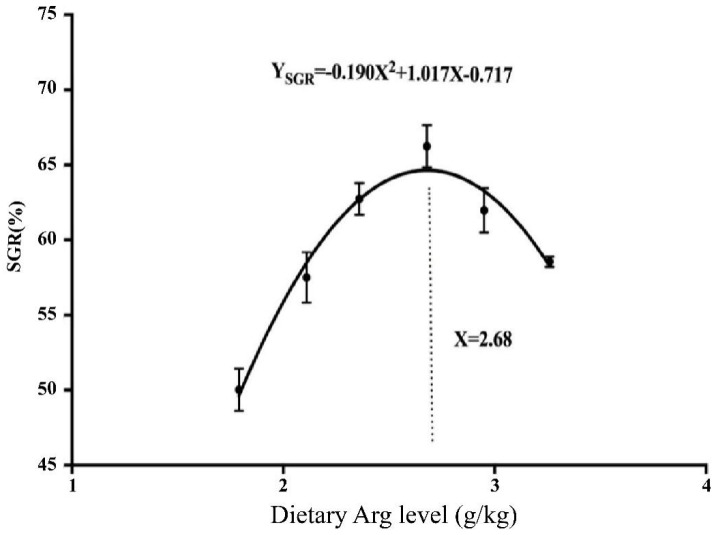
Quadratic regression analysis of specific growth rate (SGR) for yellow catfish (*Pelteobagrus fulvidraco*) fed diets containing graded levels of Arg (g/kg) for 8 weeks.

**Figure 2 biology-13-00881-f002:**
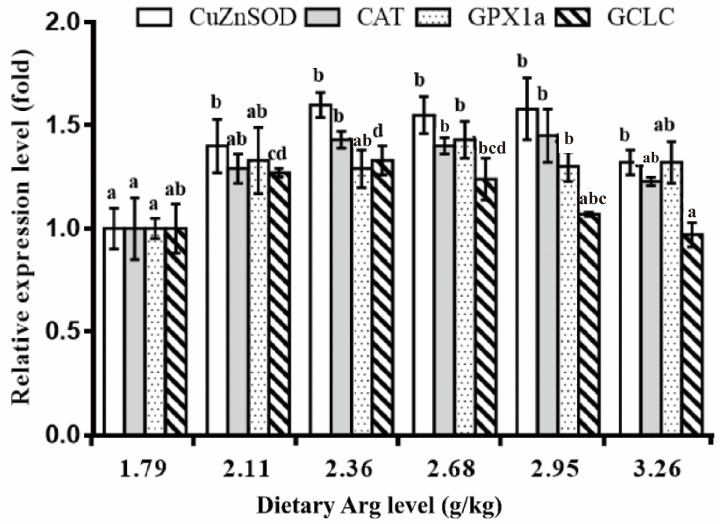
Relative expression of Copper-Zinc superoxide dismutase (CuZnSOD), catalase (CAT) and glutathione peroxidase 1a (GPx1a) and glutamate-cysteine ligase catalytic subunit (GCLC) in the intestine of yellow catfish fed diets containing graded levels of Arg (g/kg). Data represents the means of three replicate groups of yellow catfish with 12 fish per group; error bars indicate standard error. Different letters above a bar denote significant difference (*p* < 0.05).

**Figure 3 biology-13-00881-f003:**
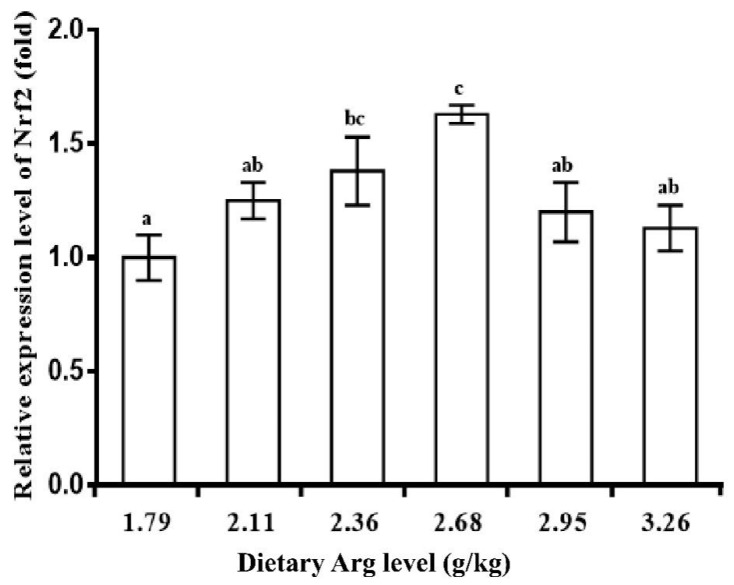
Relative expression of NF-E2-related factor 2 (Nrf2) in the intestine of yellow catfish fed diets containing graded levels of Arg (g/kg). Data represents means of three replicate groups of yellow catfish with 12 fish per group; error bars indicate standard error. Different letters above a bar denote significant difference (*p* < 0.05).

**Figure 4 biology-13-00881-f004:**
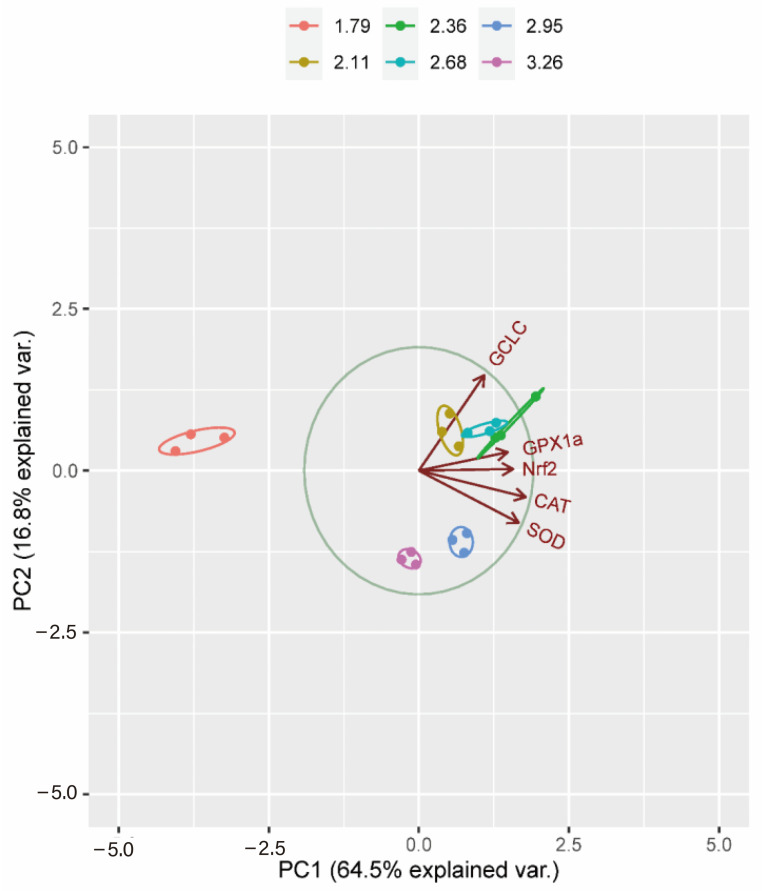
Loading plot of principal component analysis (PCA) in intestine representing measured variables and the individuals based on all the gene expressions of six graded Arg levels (g/kg) in the (PC1 × PC2) coordination plane. Ellipses in group from the six conditions (1.79 g/kg diet in red, 2.11 g/kg in yellow, 2.36 g/kg diet in green, 2.68 g/kg diet in light blue, 2.95 g/kg diet in blue and 3.26 g/kg diets in pink).

**Figure 5 biology-13-00881-f005:**
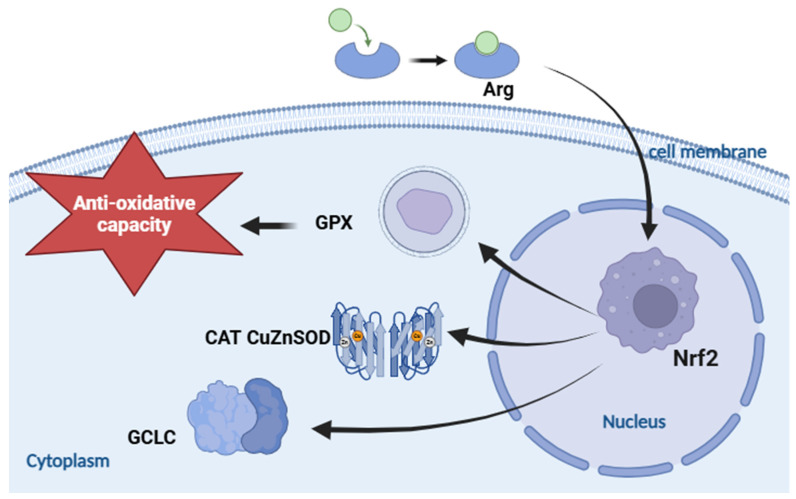
Role of Nrf2 in the regulation of intestinal metabolism in yellow catfish under Arg feed. Nrf2 can induce the downstream pathway genes including GPX, CuZnSOD, CAT, GCLC to increase the anti-oxidative capacity.

**Table 1 biology-13-00881-t001:** Composition and nutrients content of diets (g/kg).

Ingredient	Arg 0.179%	Arg 0.211%	Arg 0.236%	Arg 0.268%	Arg 0.295%	Arg 0.326%
Soybean meal	100	100	100	100	100	100
Rapeseed meal	103	103	103	103	103	103
Fishmeal	230	230	230	230	230	230
wheat flour	254	255	256	258	259	260
Corn gluten meal	210	210	210	210	210	210
Soya bean oil	20	20	20	20	20	20
Lysine	9	9	9	9	9	9
Threonine	3	3	3	3	3	3
L-Gly	21	17	13	8	4	0
L-Arg-HCL	0	3	6	9	12	15
CaH_2_PO_4_	20	20	20	20	20	20
Choline chloride	5	5	5	5	5	5
Vitamin premix/g/kg ^1^	10	10	10	10	10	10
Mineral premixg/kg ^2^	15	15	15	15	15	15
Nutrients content (%) ^3^						
Crude protein	36.8	37.2	36.8	36.8	37.2	36.9
Crude lipid	7.00	6.98	6.97	6.69	6.93	6.60
Ash	7.67	7.75	7.83	7.92	7.54	7.58
Arg	1.79	2.11	2.36	2.68	2.95	3.26

^1^ Soybean meal (COFCO Oil Qinzhou Co., Ltd., Qinzhou, China), Rapeseed meal (Chengdu Huatai Grain and Oil Ltd., Chengdu, China), Fish meal (TASA Steam dried fishmeal, Lima, Peru), Corn gluten meal (Changchun Dacheng Industry Group, Changchun, China), Wheat meal (China National Cereals, Oils and Foodstuffs Corporation, Beijing, China), Soy oil (Baker Commodities Inc., Vernon, CA, USA); ^2^ Vitamin premix (Chengdu NutriMax Animal Health Co., Ltd., Chengdu, China, IU or g/kg premix): retinyl acetate, 2,500,000 IU; cholecalciferol, 500,000 IU; α-tocopherol, 6700 IU; thiamine, 10; riboflavin, 6; pyridoxine hydrochloride, 12; nicotinic acid, 40; d-calcium pantothenate, 15; biotin, 0.25; folic acid, 0.4; inositol, 200; cyanocobalamin 0.02; menadione, 4. All ingredients were diluted with corn starch to 1 kg. ^2^ Mineral premix (g/kg premix): FeC_6_H_5_O_7_, 4.57; ZnSO_4_·7H_2_O, 9.43; MnSO_4_·H_2_O, 4.14; CuSO_4_·5H_2_O, 6.61; MgSO_4_·7H_2_O, 238.97; KI, 1.10 g; NaSeO_3_, 2.50 g; CoCl_2_·6H_2_O, 1.36. All ingredients were diluted with CaCO_3_ to 1 kg. ^3^ Crude protein, crude fat and ash were measured by using the Association of Official Analytical Chemists (AOAC, 1990) methods. Arg concentrations were measured using high performance liquid chromatography (Agilent Technologies, PaloAlto, CA, USA).

**Table 2 biology-13-00881-t002:** Primer sequences, optimal annealing temperatures (OAT), and amplification products (AP) of genes selected for analysis by real-time PCR.

Primes	Sequences	OAT (°C)	AP (bp)	Accession Number
β-actin-RTF	CCTAAAGCCAACAGGGAAAA	59.0	186 bp	EU161066
β-actin-RTR	ATGGGGCAGAGCATAACC			
CuZnSOD-RTF	ATCTGGGTAATGTGACTGCCGA	60.4	152 bp	KX455916
CuZnSOD-RTR	TTCATCATCTCCGCCCTTGC			
CAT-RTF	ACACCGATGAGGGAAACTGG	58	89 bp	KX455919
CAT-RTR	GTGGATGAAGGACGGGAACA			
GPX 1a-RTF	GTGACGACTCTGTGTCCTTG	61.0	139 bp	KY312111
GPX 1a-RTR	AACCTTCTGCTGTATCTCTTGA			
GCLC-RTF	GACAAACGGAGGAAGGAGG	58.2	161 bp	KX455918
GCLC-RTR	TCATCAGGAAAGAAGAGGGACT			
Nrf2-RTF	CGGAACAAGATGGAGAAGCC	64.0	122 bp	KX455917
Nrf2-RTR	ACAGGGAGGAATGGAGGGA			

Copper–zinc superoxide dismutase (CuZnSOD); Catalase (CAT); Glutathione peroxidase-1a (GPx1a); γ-glutamylcysteine ligase catalytic subunit (GCLC); Nuclear factor erythroid 2-related factor 2 (Nrf2).

**Table 3 biology-13-00881-t003:** Growth performance of yellow catfish fed diets with graded levels of Arg (g/kg) for 8 weeks.

Ingredient	Arg 0.179%	Arg 0.211%	Arg 0.236%	Arg 0.268%	Arg 0.295%	Arg 0.326%	*p*-Value
IBW	61.2 ± 0.17	61.2 ± 0.17	61.2 ± 0.17	61.3 ± 0.17	61.3 ± 0.17	61.3 ± 0.17	0.326
FBW	80.9 ± 0.26 ^a^	84.4 ± 0.30 ^ab^	86.9 ± 0.32 ^bc^	88.9 ± 2.45 ^c^	86.8 ± 1.74 ^bc^	85.1 ± 0.32 ^bc^	0.042
PWG	32.3 ± 0.60 ^a^	38.0 ± 0.75 ^b^	42.1 ± 0.49 ^c^	45.0 ± 0.74 ^d^	41.5 ± 0.78 ^c^	38.8 ± 0.15 ^b^	0.033
SGR	0.50 ± 0.01 ^a^	0.58 ± 0.01 ^b^	0.63 ± 0.01 ^c^	0.66 ± 0.01 ^d^	0.62 ± 0.02 ^c^	0.59 ± 0.01 ^b^	0.031
SR	100 ± 0.00	99.2 ± 0.83	98.3 ± 0.83	100 ± 0.00	97.5 ± 2.5	100 ± 0.00	0.935
FI	40.9 ± 0.89 ^a^	46.7 ± 2.03 ^ab^	48.0 ± 2.36 ^ab^	49.4 ± 2.31 ^b^	46.9 ± 2.50 ^b^	46.6 ± 2.43 ^ab^	0.047
FE	48.4 ± 0.91 ^a^	50.0 ± 0.09 ^ab^	53.6 ± 0.42 ^cd^	55.8 ± 1.22 ^d^	54.2 ± 1.06 ^d^	51.4 ± 0.62 ^bc^	0.024
PER	1.32 ± 0.03 ^a^	1.33 ± 0.03 ^ab^	1.33 ± 0.02 ^ab^	1.51 ± 0.01 ^d^	1.46 ± 0.03 ^cd^	1.41 ± 0.03 ^bc^	0.019
Regression							
Y_SGR_ = −0.190X^2^ + 1.017X − 0.717					X = 2.68	*R*^2^ = 0.933	

Values are means ± S.E. of three replicate groups with 12 fish in each group. Values within the same rows having different superscripts are significantly different (*p <* 0.05). Initial body weight (IBW, g/fish), Final body weight (FBW, g/fish), Survival rate (SR, %), Feed intake (FI, g/fish). Specific growth rate (SGR) = (ln FBW − ln IBW)/days × 100. Feed efficiency (FE) = (weight gain, g/fish)/(feed intake, g/fish) × 100. Percent weight gain (PWG) = (weight gain, g/fish)/(initial weight, g/fish) × 100. Protein efficiency ratio (PER) = weight gain (g)/protein intake (g). A significant quadratic response was observed between SGR and the Arg level, with the regression equation: YSGR = −0.190X^2^ + 1.017X − 0.717, X = 2.68, R^2^ = 0.933.

**Table 4 biology-13-00881-t004:** Whole body composition (%, wet-basis) of yellow catfish fed diets with graded levels of Arg (g/kg) 8 weeks.

Ingredient	Arg 0.179%	Arg 0.211%	Arg 0.236%	Arg 0.268%	Arg 0.295%	Arg 0.326%	*p*-Value
Moisture	67.9 ± 0.07	68.2 ± 0.42	67.5 ± 0.47	67.6 ± 0.53	68.3 ± 0.14	68.1 ± 0.80	0.658
Protein	15.6 ± 0.10 ^a^	15.6 ± 0.30 ^a^	15.6 ± 0.41 ^a^	16.5 ± 0.35 ^b^	16.5 ± 0.16 ^b^	16.0 ± 0.04 ^b^	0.039
Lipid	11.3 ± 0.26 ^a^	11.7 ± 0.23 ^ab^	12.3 ± 0.31 ^bc^	12.6 ± 0.23 ^c^	11.7 ± 0.17 ^ab^	11.4 ± 0.19 ^a^	0.042
Ash	4.58 ± 0.16	4.51 ± 0.13	4.46 ± 0.15	4.20 ± 0.10	4.43 ± 0.03	4.44 ± 0.07	0.527
PPV	14.2 ± 0.14 ^a^	15.4 ± 0.57 ^a^	14.9 ± 0.82 ^a^	22.8 ± 1.03 ^c^	19.5 ± 0.66 ^b^	18.1 ± 0.60 ^b^	0.016
LPV	96.0 ± 6.02 ^a^	115 ± 7.92 ^ab^	131 ± 9.47^bc^	144 ± 7.43 ^c^	113 ± 6.03 ^ab^	112 ± 3.42 ^ab^	0.021
APV	33.9 ± 1.61	35.6 ± 0.51	33.9 ± 0.80	35.0 ± 0.83	34.5 ± 0.44	33.6 ± 0.93	0.834

Values are means ± S.E. of three replicate groups with 12 fish in each group. Values within the same rows having different superscripts are significantly different (*p <* 0.05). Protein production value (PPV) = fish protein gain (g)/total protein intake (g) × 100. Lipid production value (LPV) = fish lipid gain (g)/total lipid intake (g) × 100. Ash production value (APV) = fish ash gain (g)/total ash intake (g) × 100.

**Table 5 biology-13-00881-t005:** Stomach protein content (SPC), pancreas protein content (PPC), intestosomatic index (ISI), relative gut length (RGL), whole intestinal protein content (IPC), hepatosomatic index (HSI), and liver protein content (LPC) of yellow catfish fed diets with graded levels of Arg (g/kg) for 8 weeks.

Ingredient	Arg 0.179%	Arg 0.211%	Arg 0.236%	Arg 0.268%	Arg 0.295%	Arg 0.326%	*p*-Value
Stomach							
SPC	3.14 ± 0.12	3.32 ± 0.24	3.58 ± 0.22	3.50 ± 0.10	3.19 ± 0.05	3.07 ± 0.05	0.237
Pancreas							
PPC	3.23 ± 0.09 ^a^	3.70 ± 0.13 ^bc^	3.86 ± 0.06 ^c^	4.26 ± 0.12 ^d^	3.82 ± 0.16 ^c^	3.46 ± 0.07 ^ab^	0.018
Intestine							
ISI	0.99 ± 0.03	1.05 ± 0.02	1.03 ± 0.03	1.06 ± 0.04	1.07 ± 0.03	1.08 ± 0.04	0.569
RGL	0.60 ± 0.02 ^a^	0.68 ± 0.01 ^b^	0.63 ± 0.02 ^ab^	0.65 ± 0.02 ^ab^	0.64 ± 0.01 ^ab^	0.64 ± 0.02 ^ab^	0.035
IPC	2.85 ± 0.21 ^a^	2.92 ± 0.12 ^ab^	3.29 ± 0.10 ^b^	3.17 ± 0.12 ^ab^	3.16 ± 0.09 ^ab^	2.90 ± 0.09 ^ab^	0.027
Liver							
HSI	1.37 ± 0.04	1.37 ± 0.05	1.48 ± 0.07	1.42 ± 0.04	1.45 ± 0.03	1.43 ± 0.05	0.663
LPC	9.41 ± 0.21 ^a^	10.1 ± 0.13 ^a^	10.7 ± 0.57 ^ab^	11.1 ± 0.13 ^b^	9.81 ± 0.12 ^a^	9.21 ± 0.32 ^a^	0.047

Values are means ± S.E. of three replicate groups with 12 fish in each group. Values within the same rows having different superscripts are significantly different (*p <* 0.05). HSI = wet liver weight (g)/wet body weight (g) × 100; ISI = wet intestine weight (g)/wet body weight (g) × 100. RGL = whole intestine length (cm)/body length (cm) × 100

**Table 6 biology-13-00881-t006:** The digestive enzyme activities (U/g tissue) of yellow catfish fed diets with graded levels of Arg (g/kg) for 8 weeks.

Ingredient	Arg 0.179%	Arg 0.211%	Arg 0.236%	Arg 0.268%	Arg 0.295%	Arg 0.326%	*p*-Value
Stomach							
Pepsin	967 ± 57.2 ^a^	1057 ± 26.2 ^ab^	1519 ± 26.8 ^c^	1129 ± 22.4 ^b^	1020 ± 65.5 ^ab^	1042 ± 29.0 ^ab^	0.023
Pancreas							
Trypsin	20.6 ± 0.4 ^a^	31.8 ± 2.1 ^bc^	36.3 ± 3.3 ^bcd^	42.0 ± 1.7 ^d^	35.5 ± 1.6 ^bc^	30.0 ± 1.4 ^b^	<0.001
Chymotrypsin	27.6 ± 1.2 ^a^	28.8 ± 1.6 ^a^	33.6 ± 1.2 ^b^	46.8 ± 3.6 ^d^	36.0 ± 1.3 ^c^	26.4 ± 0.6 ^a^	0.019
Lipase	6.42 ± 0.48 ^a^	8.80 ± 0.22 ^ab^	9.59 ± 0.59 ^b^	13.3 ± 0.86 ^c^	9.84 ± 0.15 ^b^	6.80 ± 0.23 ^a^	<0.001
Amylase	1617 ± 33 ^a^	2099 ± 84 ^ab^	2255 ± 75 ^b^	2121 ± 34 ^ab^	2016 ± 34 ^ab^	2028 ± 90 ^ab^	0.025
Intestine							
Trypsin	128 ± 3.3 ^a^	145 ± 2.8 ^b^	190 ± 6.8 ^d^	199 ± 8.6 ^d^	186 ± 2.8 ^d^	162 ± 3.9 ^c^	0.033
Chymotrypsin	60.1 ± 2.8 ^a^	66.1 ± 6.0 ^a^	118 ± 6.0 ^c^	90.1 ± 2.1 ^b^	92.5 ± 3.2 ^b^	93.7 ± 3.6 ^b^	<0.001
Lipase	2.10 ± 0.10 ^a^	2.13 ± 0.11 ^ab^	2.35 ± 0.09 ^ab^	2.43 ± 0.09 ^bc^	2.66 ± 0.12 ^c^	2.27 ± 0.01 ^ab^	0.046
Amylase	90.0 ± 9.3	93.7 ± 1.6	107 ± 9.9	105 ± 5.2	105 ± 7.8	90.9 ± 10.1	0.095
Regression							
Y_Intestine Trypsin_ = −89.696X^2^ + 483.244X − 457.516					X = 2.69	*R*^2^ = 0.819	
Y_Pancreas Trypsin_ = −26.725X^2^ + 141.121X − 146.711					X = 2.64	*R*^2^ = 0.823	

Values are means ± S.E. of three replicate groups with 12 fish in each group, while quadratic regressions were run with the triplicate data points. Values within the same rows having different superscripts are significantly different (*p <* 0.05).

**Table 7 biology-13-00881-t007:** The activities of alkaline phosphatase (AKP, U/g tissue), Na^+^, K^+^-ATPase (NKA, U/g tissue), γ-glutamyl transpeptidase (γ-GT, U/g tissue) and creatine kinase (CK, U/g tissue) in proximal intestine (PI) and distal intestine (DI) of yellow catfish fed diets with graded levels of Arg (g/kg) for 8 weeks.

Ingredient	Arg 0.179%	Arg 0.211%	Arg 0.236%	Arg 0.268%	Arg 0.295%	Arg 0.326%	*p*-Value
PI							
γ-GT	54.9 ± 0.9 ^a^	54.1 ± 1.5 ^a^	62.0 ± 0.9 ^b^	60.3 ± 1.2 ^b^	53.4 ± 0.5 ^a^	55.3 ± 0.5 ^a^	0.046
CK	54.5 ± 2.4 ^ab^	53.5 ± 0.8 ^ab^	58.9 ± 2.4 ^abc^	60.8 ± 2.1 ^bc^	62.1 ± 2.6 ^c^	52.9 ± 1.8 ^a^	0.037
NKA	190 ± 15.8 ^ab^	198 ± 8.6 ^ab^	217 ± 8.4 ^ab^	231 ± 10.7 ^b^	198 ± 10.8 ^ab^	173 ± 7.6 ^a^	0.029
AKP	21.3 ± 2.5 ^b^	22.8 ± 1.2 ^b^	30.6 ± 2.5 ^c^	24.7 ± 3.0 ^bc^	18.5 ± 0.4 ^b^	14.7 ± 0.1 ^a^	<0.001
DI							
γ-GT	11.1 ± 1.1 ^a^	15.5 ± 0.8 ^b^	18.0 ± 1.1 ^b^	25.4 ± 1.7 ^c^	18.1 ± 1.4 ^b^	15.3 ± 1.3 ^b^	<0.001
CK	51.3 ± 2.1 ^a^	60.1 ± 3.0 ^b^	84.5 ± 0.9 ^c^	61.0 ± 2.8 ^b^	57.3 ± 1.3 ^ab^	55.2 ± 2.3 ^ab^	0.017
NKA	191 ± 5.2 ^a^	227 ± 15.6 ^bc^	239 ± 8.1 ^bc^	256 ± 9.1 ^c^	236 ± 7.7 ^bc^	215 ± 3.5 ^ab^	0.035
AKP	15.2 ± 0.09 ^ab^	18.9 ± 0.66 ^b^	29.7 ± 0.61 ^d^	25.3 ± 1.3 ^c^	17.9 ± 0.42 ^b^	13.1 ± 1.0 ^a^	0.016
Regression							
Y_DI AKP_ = −23.604X^2^ + 117.556X − 120.526					X = 2.49	*R*^2^ = 0.751	

Values are means ± S.E. of three replicate groups with 12 fish in each group, while quadratic regression was run with the triplicate data points. Values within the same rows having different superscripts are significantly different (*p <* 0.05).

**Table 8 biology-13-00881-t008:** The effect of dietary Arg on intestinal antioxidant activities including malondialdehyde (MDA), protein carbonyl (PC), total Superoxide Dismutase (T-SOD), catalase (CAT), glutathione-S-transferase (GST), glutathione peroxidase (GPx), glutathione reductase (GR), total glutathione (T-GSH), anti-superoxide anion (ASA) and anti-hydroxyl radical (AHR) in yellow catfish fed with graded levels of Arg (g/kg) for 8 weeks.

Ingredient	Arg 0.179%	Arg 0.211%	Arg 0.236%	Arg 0.268%	Arg 0.295%	Arg 0.326%	*p*-Value
MDA	95.1 ± 5.4 ^e^	86.4 ± 6.9 ^cd^	77.7 ± 1.2 ^bc^	59.3 ± 2.4 ^a^	70.4 ± 2.1 ^ab^	76.5 ± 2.5 ^bc^	<0.001
PC	140 ± 3.8 ^bc^	128 ± 3.8 ^bc^	121 ± 13.7 ^b^	83.3 ± 7.6 ^a^	125 ± 17.4 ^b^	163 ± 10.0 ^c^	<0.001
T-SOD	79.2 ± 6.6 ^a^	92.0 ± 4.7 ^ab^	101.5 ± 4.9 ^b^	105.6 ± 2.3 ^b^	102.3 ± 1.9 ^b^	96.6 ± 2.5 ^b^	0.021
CAT	54.0 ± 3.6 ^a^	69.0 ± 3.9 ^b^	72.3 ± 4.7 ^b^	75.5 ± 8.1 ^b^	54.9 ± 3.0 ^a^	44.3 ± 2.1 ^a^	0.006
GST	126 ± 7.1 ^a^	166 ± 7.9 ^b^	176 ± 7.8^b^	213 ± 14.2 ^c^	187 ± 14.1 ^bc^	160 ± 7.6 ^b^	<0.001
GPX	5393 ± 210 ^a^	6190 ± 261 ^b^	6270 ± 174 ^b^	6040 ± 220 ^b^	6088 ± 174 ^b^	5997 ± 124 ^ab^	0.003
GR	41.2 ± 2.4 ^a^	43.8 ± 3.6 ^ab^	52.1 ± 5.0 ^bc^	58.9 ± 2.3 ^c^	52.6 ± 2.2 ^bc^	47.8 ± 1.7 ^ab^	<0.001
T-GSH	0.76 ± 0.04 ^ab^	0.83 ± 0.02 ^bc^	0.90 ± 0.04 ^c^	0.83 ± 0.02 ^bc^	0.76 ± 0.03 ^ab^	0.72 ± 0.03 ^a^	0.029
ASA	261 ± 7.7 ^a^	277 ± 7.7 ^ab^	299 ± 11.7 ^bc^	325 ±11.6 ^c^	312 ± 6.5 ^bc^	311 ± 6.1 ^bc^	0.006
AHR	218 ± 6.6 ^a^	259 ± 19.7 ^ab^	252 ± 18.5 ^ab^	288 ± 8.5 ^b^	222 ± 7.8 ^a^	226 ± 12.0 ^a^	0.027

Values are means ± S.E. of three replicate groups with 12 fish in each group. Values within the same rows having different superscripts are significantly different (*p <* 0.05).

## Data Availability

Not available.

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
