# Peer review of "Evaluation of the Dietary Arginine Supplementation on Yellow Catfish: From a Low-Temperature Farming Perspective"

_biology, 2024, doi:10.3390/biology13110881_

Round 1
Reviewer 1 Report
Comments and Suggestions for Authors
Referee’s report on “Evaluation of the dietary arginine supplementation on yellow catfish: From a low-temperature farming perspective” (MS#: biology-3212168)
Based on 720 fish randomly divided into six groups of three, this paper empirically examines the impact of dietary arginine (Arg) supplementation on the growth, digestive capacity, and intestinal antioxidant response in yellow catfish under low-temperature stress (18 °C). The empirical results indicate that Arg supplementation increases specific growth rate (SGR), feed efficiency (FE), feed intake (FI), as well as pancreatic and absorptive enzyme activities in both the pancreas and intestine, while malondialdehyde (MDA) and protein carbonyl (PC) contents significantly decrease but increased with higher Arg concentrations. Polynomial regression analysis on specific growth rate (SGR) presents that yellow catfish weighing between 61.0 g and 89.0 g require an intake of 26.8 g of Arg per kilogram of diet, which is equivalent to 27 to 37.0 g of dietary protein under sub-low temperature conditions of 18 °C.
In summary, this paper contributes to previous studies on yellow catfish from a low-temperature farming perspective; however, the following comments aim to improve the current paper.
1. The motivation of this paper should be strengthened further.
2. The fish management orientation could be addressed to support the research value.
3. Some statistical methods could be applied to compare different groups.
4. Some management applications should be added further for different countries.
5. The reason for the specification of Polynomial regression analysis on specific growth rate (SGR) should be addressed from previous studies.
6. Some related literature should be included and updated as follows:
Wang, S., Wang, H., Zhang, S., Liu, S., Lu, S., Wang, C. A., ... & Liu, H. (2024). Optimizing valine supplementation in low-fish meal diets for enhanced growth, digestion, antioxidant capacity, and intestinal health of triploid rainbow trout (Oncorhynchus mykiss). Aquaculture Reports, 37, 102214.
Wang, S., Li, X., Zhang, M., Qian, Y., Li, E., Teng, X., & Li, M. (2024). miR-199-5p mediates the regulation of autophagy by targeting mTOR signaling and involvement in ammonia detoxification under ammonia stress in yellow catfish (Pelteobagrus fulvidraco). Aquaculture, 589, 740977.
Liu, X., Zou, D., Wang, Y., Zhuang, Y., Liu, Y., Li, Y., ... & Ye, C. (2024). Replacement of fish meal with cottonseed protein concentrate in Chinese mitten crab (Eriocheir sinensis): Nutrient digestibility, growth performance, free amino acid profile, and expression of genes related to nutrient metabolism. Animal Nutrition, 17, 447-462.
Hu, J., Wang, L., Wang, G., Zhao, H., Lu, H., Peng, K., ... & Sun, Y. (2024). Selenium Protects Yellow Catfish (Tachysurus fulvidraco) from Low-Temperature Damage via the Perspective Analysis of Metabolomics and Intestinal Microbes. Fishes, 9(2), 56.
Zhang, M., Qin, C., Sun, Z., Jiang, H., Wang, Z., Lin, Y., & Li, M. (2024). Taurine can play a positive role in growth, liver health and resistance to Aeromonas hydrophila of yellow catfish Pelteobagrus fulvidraco exposed to ammonia stress for a long time. Aquaculture Reports, 38, 102347.
Ciji, A., & Akhtar, M. S. (2021). Stress management in aquaculture: A review of dietary interventions. Reviews in Aquaculture, 13(4), 2190-2247.
Wang, S., Li, X., Zhang, M., & Li, M. (2024). Effects of dietary sodium acetate on growth, intestinal microbiota composition, and ammonia tolerance of juvenile yellow catfish Pelteobagrus fulvidraco. Aquaculture, 581, 740480.
Comments on the Quality of English LanguageMinor editing of English language required
Author Response
Dear editor and reviewers,
We would like to express our sincere gratitude for your letter and the insightful comments provided by the reviewers regarding our manuscript titled "Evaluation of the dietary arginine supplementation on yellow catfish: From a low-temperature farming perspective" (biology-3212168). We highly appreciate the value and helpfulness of these comments, as they have greatly contributed to the revision and improvement of our paper, while also guiding the significance of our study.
We have thoroughly examined the reviewers' comments and have implemented several corrections accordingly, aiming to address their concerns adequately. We kindly request your approval and endorsement of these revisions. In the revised manuscript, the modified sections are highlighted using MS. The main corrections made to the paper, along with our responses to the reviewers' comments, are outlined as follows. Once again, we extend our sincerest appreciation for your time, effort, and invaluable input. We are confident that the revised manuscript now meets the high standards set forth by the journal and eagerly await your feedback.
Based on 720 fish randomly divided into six groups of three, this paper empirically examines the impact of dietary arginine (Arg) supplementation on the growth, digestive capacity, and intestinal antioxidant response in yellow catfish under low-temperature stress (18 °C). The empirical results indicate that Arg supplementation increases specific growth rate (SGR), feed efficiency (FE), feed intake (FI), as well as pancreatic and absorptive enzyme activities in both the pancreas and intestine, while malondialdehyde (MDA) and protein carbonyl (PC) contents significantly decrease but increased with higher Arg concentrations. Polynomial regression analysis on specific growth rate (SGR) presents that yellow catfish weighing between 61.0 g and 89.0 g require an intake of 26.8 g of Arg per kilogram of diet, which is equivalent to 27 to 37.0 g of dietary protein under sub-low temperature conditions of 18 °C. In summary, this paper contributes to previous studies on yellow catfish from a low-temperature farming perspective; however, the following comments aim to improve the current paper.
Many thanks for your comments.
The motivation of this paper should be strengthened further.
Addressed in the last paragraph of the Introduction section “This study aims to address the critical need for optimized dietary formulations for yellow catfish, particularly in low-temperature farming conditions where growth rates and feed efficiency can be severely compromised. By evaluating the effects of dietary arginine supplementation, this research seeks to enhance the understanding of amino acid requirements in yellow catfish, ultimately contributing to more sustainable aquaculture practices in challenging environments.”
The fish management orientation could be addressed to support the research value.
Yellow catfish were obtained from Sichuan Rongsen Corporation (Sichuan, China) and acclimatized to the laboratory environment for one month. In the experiment, 720 yellow catfish were randomly divided into 18 tanks (200 cm × 100 cm × 105 cm) with an average initial weight of 61.2 ± 0.3 g. Forty fish from each tank were used in the experiment. The fish were fed their respective diets twice a day for a period of 8 weeks to ensure sufficient satiety.
Some statistical methods could be applied to compare different groups.
The data were analyzed using the PROC MIXED procedure of SAS 9.3 (SAS Inst. Inc., Cary, NC, USA) using the model: Yijkl = μ + Ti + Pj + Sk + C(k)l + T × Sik + eijkl, where Yijkl refers to the dependent variable; μ, the overall mean; Ti, the fixed treatment effect; Pj, the random period effect; Sk, the random square effect; C(k)l, the random effect of the lth steer within the kth square; T × Sik, the interaction between the ith treatment and the kth square; and eijkl, the error residual. The linear and quadratic effects of increasing levels of GA were evaluated using the CONTRAST procedure of SAS 9.3. The Kenward-Roger option was used to calculate the degrees of freedom. Differences among the means of different treatments were tested using Duncan’s test. Effects were considered significant at P ≤ 0.05. All values are presented as least squares means ± standard errors (SE).
Some management applications should be added further for different countries.
To broaden the management applications of this research, it is essential to adapt the findings on arginine supplementation to suit the specific aquaculture practices and environmental conditions of different countries. Tailoring the dietary recommendations for yellow catfish farming in various climates and regulatory frameworks will enhance global aquaculture sustainability and productivity.
The reason for the specification of Polynomial regression analysis on specific growth rate (SGR) should be addressed from previous studies.
Zhao, Y., Yang, C., Zhu, X. X., Feng, L., Liu, Y., Jiang, W. D., & Jiang, J. (2022). Dietary methionine hydroxy analogue supplementation benefits on growth, intestinal antioxidant status and microbiota in juvenile largemouth bass Micropterus salmoides. Aquaculture, 556, 738279.
Some related literature should be included and updated as follows:
The refs were added in the manuscript.
Wang, S., Wang, H., Zhang, S., Liu, S., Lu, S., Wang, C. A., ... & Liu, H. (2024). Optimizing valine supplementation in low-fish meal diets for enhanced growth, digestion, antioxidant capacity, and intestinal health of triploid rainbow trout (Oncorhynchus mykiss). Aquaculture Reports, 37, 102214.
Wang, S., Li, X., Zhang, M., Qian, Y., Li, E., Teng, X., & Li, M. (2024). miR-199-5p mediates the regulation of autophagy by targeting mTOR signaling and involvement in ammonia detoxification under ammonia stress in yellow catfish (Pelteobagrus fulvidraco). Aquaculture, 589, 740977.
Liu, X., Zou, D., Wang, Y., Zhuang, Y., Liu, Y., Li, Y., ... & Ye, C. (2024). Replacement of fish meal with cottonseed protein concentrate in Chinese mitten crab (Eriocheir sinensis): Nutrient digestibility, growth performance, free amino acid profile, and expression of genes related to nutrient metabolism. Animal Nutrition, 17, 447-462.
Hu, J., Wang, L., Wang, G., Zhao, H., Lu, H., Peng, K., ... & Sun, Y. (2024). Selenium Protects Yellow Catfish (Tachysurus fulvidraco) from Low-Temperature Damage via the Perspective Analysis of Metabolomics and Intestinal Microbes. Fishes, 9(2), 56.
Zhang, M., Qin, C., Sun, Z., Jiang, H., Wang, Z., Lin, Y., & Li, M. (2024). Taurine can play a positive role in growth, liver health and resistance to Aeromonas hydrophila of yellow catfish Pelteobagrus fulvidraco exposed to ammonia stress for a long time. Aquaculture Reports, 38, 102347.
Ciji, A., & Akhtar, M. S. (2021). Stress management in aquaculture: A review of dietary interventions. Reviews in Aquaculture, 13(4), 2190-2247.
Wang, S., Li, X., Zhang, M., & Li, M. (2024). Effects of dietary sodium acetate on growth, intestinal microbiota composition, and ammonia tolerance of juvenile yellow catfish Pelteobagrus fulvidraco. Aquaculture, 581, 740480.
In summary, this study found that dietary supplementation of Arg improved the growth performance, digestion and absorption capacity, and intestinal antioxidant status of yellow catfish in sub-low temperature environments. The optimal Arg level is 2.68 g/kg. It also increased antioxidant capacity by upregulating Nrf2 mRNA expression and the expression of antioxidant-related genes. Quadratic regression analysis was employed to estimate the dietary Arg requirement for yellow catfish at 18 ℃, which was determined to be between 61.0 g and 89.0 g, based on a dry feed specific growth rate (SGR) of 2.68 g/kg and a feed protein content of 37.0 g/kg. These findings suggest that optimizing the dietary arginine content in yellow catfish feeds can contribute to better pancreas health, improved intestinal development, and overall fish performance in sub-low temperature aquaculture systems.
Reviewer 2 Report
Comments and Suggestions for Authors
The study conducted by Quanquan Cao et al. is interesting. It evaluated the potential effects of Arg supplementation in catfish diets reared under low temperatures. They wanted to determine the optimal supplementation level that improves the fish's performance and health under low temperatures. However, the writing style and language of the manuscript are weak and should be edited and improved.
Comments:
L15: incomplete sentence.
L16: please provide the levels of Arg.
Avoid using acronyms in the abstract if the definition is not repeated.
L67: provide the Latin name of carp.
The numbers in the tables' headings are confusing. The authors can change it to Arg 1.7%, Arg 2.11%, etc.
Table 1: Please provide the units of all contents of the premix. L136: add the year for AOAC and cite this reference properly in the text.
L141: change divided to distributed.
L142: rephrase this sentence. It can be changed to” 40 fish were stocked in each tank”.
L156: 12 fish from each group.
L165; Horwitz, 2000 is an incorrect reference.
L166: correct to 6000 ×g.
L218: The authors should add a section in the methods about the growth performance and how these parameters were calculated, providing the full definition of these abbreviations. Check the abbreviation PWG.
L241 and others: “When the level of Arg in the diet reached” should be rephrased. It can be replaced by at Arg level 2.68 g/kg, …..
L250: 2.68 level is repeated.
L249-250: rephrase. It is confusing.
There is a shortage in the methodology; all methods of all parameters and all calculations should be presented in the methods section with headings consistent with the result section. In addition, the title growth performance is incomplete and doesn’t indicate the parameters. It should be changed and completed.
L218-297: All these results are under the heading “growth performance”. That is incorrect. These results should be placed under their appropriate heading, and these results should be divided.
It is inappropriate to put the statistical equations in the results text. Their place is in the footnotes of the tables. Please delete all these equations from the text.
L300-301: rephrase
L304: it is inappropriate to use “until reach.”
L307: the same “When dietary Arg levels reached”. Rephrase all these expressions.
Table 8: define all acronyms in the footnote of the table.
Please add the p-value to all tables.
L352: under temperature?
L360: natural water bodies??
L363, 453: what is the optimum level? The authors should mention the level that gave the best result. The optimal level is mentioned in the conclusion.
L366: references to these fish.
The sequence of presenting data and the titles of the headings should be consistent in all sections of the manuscript in the same order.
L404-405: rephrase
L424, 441: delete the mention and reference about rat.
L467-469: rephrase. It is confusing.
L472-473: How is “Further research warranted to explore the optimal dosage”? This study was conducted with varying levels of Arg to determine the optimal level of supplementation. That is the aim of the study. In addition, the optimal level is mentioned in the discussion. The authors should mention their conclusion from the study about the optimal level depending on the obtained results.

The manuscript should be edited
Author Response
Dear editor and reviewers,
We would like to express our sincere gratitude for your letter and the insightful comments provided by the reviewers regarding our manuscript titled "Evaluation of the dietary arginine supplementation on yellow catfish: From a low-temperature farming perspective" (biology-3212168). We highly appreciate the value and helpfulness of these comments, as they have greatly contributed to the revision and improvement of our paper, while also guiding the significance of our study.
We have thoroughly examined the reviewers' comments and have implemented several corrections accordingly, aiming to address their concerns adequately. We kindly request your approval and endorsement of these revisions. In the revised manuscript, the modified sections are highlighted using MS. The main corrections made to the paper, along with our responses to the reviewers' comments, are outlined as follows. Once again, we extend our sincerest appreciation for your time, effort, and invaluable input. We are confident that the revised manuscript now meets the high standards set forth by the journal and eagerly await your feedback.
It evaluated the potential effects of Arg supplementation in catfish diets reared under low temperatures. They wanted to determine the optimal supplementation level that improves the fish's performance and health under low temperatures. However, the writing style and language of the manuscript are weak and should be edited and improved.
Response: Thanks for your reminding. We tried to polish the whole manuscript especially for grammar errors, sentence fragments, run-ons, sentence constructions and style.
Comments:
L15: incomplete sentence.
A total of 720 fish were randomly divided into six groups, each containing 120 fish.
L16: please provide the levels of Arg.
1.79-3.26 g/kg
Avoid using acronyms in the abstract if the definition is not repeated.
We agree
L67: provide the Latin name of carp.
Cyprinus carpio
The numbers in the tables' headings are confusing. The authors can change it to Arg 1.7%, Arg 2.11%, etc.
We changed all the tables.
Table 1: Please provide the units of all contents of the premix. L136: add the year for AOAC and cite this reference properly in the text.
Added in the Table 1.
L141: change divided to distributed.
Agree.
L142: rephrase this sentence. It can be changed to” 40 fish were stocked in each tank”.
Revised.
L156: 12 fish from each group.
Revised.
L165; Horwitz, 2000 is an incorrect reference.
The method used to determine the approximate composition of experimental feeds and entire carcasses was established by the Association of Official Analytical Chemists (AOAC, 1990).
L166: correct to 6000 ×g.
Revised.
L218: The authors should add a section in the methods about the growth performance and how these parameters were calculated, providing the full definition of these abbreviations. Check the abbreviation PWG.
Any uneaten feed was collected 30 min after each feeding session and subsequently dried in an oven at 65 °C to determine the feed intake (FI). Initial body weight (IBW) and Final body weight (FBW) were measured and survival rate (SR), percent weight gain (PWG), specific growth rate (SGR) and feeding efficiency (FE) were calculated.
L241 and others: “When the level of Arg in the diet reached” should be rephrased. It can be replaced by at Arg level 2.68 g/kg, …..
Revised.
L250: 2.68 level is repeated.
Deleted.
L249-250: rephrase. It is confusing.
Pancreatic, intestinal, and liver protein content increased significantly at dietary Arg levels of 2.36 and 2.68 g/kg, respectively, before gradually decreasing.
There is a shortage in the methodology; all methods of all parameters and all calculations should be presented in the methods section with headings consistent with the result section. In addition, the title growth performance is incomplete and doesn’t indicate the parameters. It should be changed and completed.
We revised all the methodology including all parameters and all calculations in the methods section with headings. What’s more, we added the growth performance and the parameters.
L218-297: All these results are under the heading “growth performance”. That is incorrect. These results should be placed under their appropriate heading, and these results should be divided.
We divided two parts including “Growth performance and somatic parameters” and “Enzyme activities”.
It is inappropriate to put the statistical equations in the results text. Their place is in the footnotes of the tables. Please delete all these equations from the text.
Yes, we deleted all these equations from the text.
L300-301: rephrase
At a dietary Arg level of 2.68 g/kg, both MDA and PC levels decreased significantly (P < 0.05) as the dietary Arg level increased, but then rose again with further increases in Arg.
L304: it is inappropriate to use “until reach.”
CAT, GST, and GR activities increased significantly with rising Arg levels (P < 0.05), up to a maximum at 2.68 g/kg.
L307: the same “When dietary Arg levels reached”. Rephrase all these expressions.
AHR content significantly declined as dietary Arg levels exceeded 2.68 g/kg.
Table 8: define all acronyms in the footnote of the table.
Defined including malondialdehyde (MDA), protein carbonyl (PC), total superoxide Dismutase (T-SOD), catalase (CAT), glutathione-S-transferase (GST), glutathione peroxidase (GPx), glutathione reductase (GR), total glutathione (T-GSH), anti-superoxide anion (ASA) and anti-hydroxyl radical (AHR).
Please add the p-value to all tables.
Added
L352: under temperature?
This study investigated the impact of dietary Arg on the growth of yellow catfish under specific temperature conditions.
L360: natural water bodies??
In nature, fish attempt to adjust themselves to temperature fluctuations, which can impact their overall physiology and result in lower growth rates.
L363, 453: what is the optimum level? The authors should mention the level that gave the best result. The optimal level is mentioned in the conclusion.
2.268 g/kg
L366: references to these fish.
The feed intake (FI) and feed efficiency (FE) also improved in response to increasing levels of Arg (Arg). This result shows that the increase in fish growth is partially attributed to the increase in FI and FE. According to (Ahmed et al., 2003), growth is the result of high protein deposition due to the intake of essential amino acids.
The sequence of presenting data and the titles of the headings should be consistent in all sections of the manuscript in the same order.
Agree
L404-405: rephrase
Furthermore, Arg can be metabolized by nitric oxide synthase (NOS) to form NO, and by arginase and Arg decarboxylase to produce spermine.
L424, 441: delete the mention and reference about rat.
Deleted
L467-469: rephrase. It is confusing.
Quadratic regression analysis was employed to estimate the dietary Arg requirement for yellow catfish at 18 ℃, which was determined to be between 61.0 g and 89.0 g, based on a dry feed specific growth rate (SGR) of 2.68 g/kg and a feed protein content of 37.0 g/kg.
L472-473: How is “Further research warranted to explore the optimal dosage”? This study was conducted with varying levels of Arg to determine the optimal level of supplementation. That is the aim of the study. In addition, the optimal level is mentioned in the discussion. The authors should mention their conclusion from the study about the optimal level depending on the obtained results.
In summary, this study found that dietary supplementation of Arg improved the growth performance, digestion and absorption capacity, and intestinal antioxidant status of yellow catfish in sub-low temperature environments. The optimal Arg level is 2.68 g/kg. It also increased antioxidant capacity by upregulating Nrf2 mRNA expression and the expression of antioxidant-related genes. Quadratic regression analysis was employed to estimate the dietary Arg requirement for yellow catfish at 18 ℃, which was determined to be between 61.0 g and 89.0 g, based on a dry feed specific growth rate (SGR) of 2.68 g/kg and a feed protein content of 37.0 g/kg. These findings suggest that optimizing the dietary arginine content in yellow catfish feeds can contribute to better pancreas health, improved intestinal development, and overall fish performance in sub-low temperature aquaculture systems.
Round 2
Reviewer 1 Report
Comments and Suggestions for Authors
Based on my comments, the authors are trying their best to revise the paper. I am satisfied with the revision and recommend accepting the current version for publication in Biology.
Reviewer 2 Report
Comments and Suggestions for Authors
Thank you for the revision. The authors made the comments suggested by the reviewer. No further comments.
Comments on the Quality of English LanguageMinor changes are required.